# Decreased Peak Left Atrial Longitudinal Strain Is Associated with Persistent Pulmonary Hypertension Associated with Left Heart Disease

**DOI:** 10.3390/jcm11123510

**Published:** 2022-06-18

**Authors:** Ju-Hee Lee, Jae-Hyeong Park, In-Chang Hwang, Jin Joo Park, Jun-Bean Park

**Affiliations:** 1Division of Cardiology, Department of Internal Medicine, Chungbuk National University Hospital, Chungbuk National University College of Medicine, Cheongju 28644, Korea; juheelee.md@gmail.com; 2Department of Cardiology in Internal Medicine, Chungnam National University Hospital, Chungnam National University College of Medicine, Daejeon 35015, Korea; 3Department of Cardiology, Cardiovascular Center, Seoul National University College of Medicine, Seoul National University Bundang Hospital, Seongnam 13620, Korea; inchang.hwang@gmail.com (I.-C.H.); jinjooparkmd@gmail.com (J.J.P.); 4Department of Internal Medicine, Seoul National University College of Medicine, Seoul National University Hospital, Seoul 03080, Korea; nanumy1@gmail.com

**Keywords:** heart failure, left atrium, strain echocardiography, pulmonary hypertension

## Abstract

Pulmonary hypertension (PH) associated with left heart disease (PH-LHD) is the most common form of PH and has significantly higher morbidity and mortality. We estimated the prevalence of PH-LHD on the follow-up echocardiography and the role of left atrial (LA) function in PH-LHD. From the STRATS-AHF registry composed of 4312 acute heart failure (HF) patients, we analyzed peak atrial longitudinal strain (PALS) in 1729 patients with follow-up echocardiographic examinations during mean 18.1 ± 13.5 months. PH was determined by the maximal velocity of tricuspid regurgitation (TR Vmax ≥ 3.4 m/s). Persistent PH was found in 373 patients (21.6%). The PH-LHD group was significantly older, and the prevalence of atrial fibrillation (AF), hypertension, diabetes, and heart failure with preserved ejection fraction were higher compared with the no PH-LHD group. Baseline left ventricular end-systolic volume and PALS were lower, and LA diameter, mitral E/E’ ratio, and TR Vmax were higher in the PH-LHD group. In the multivariate analysis, PALS (HR = 1.024, *p* = 0.040) was a significant variable after adjustment of LA diameter and mitral E/E’. A decreased PALS of <12.5% was the best cutoff value in the prediction of persistent PH-LHD (AUC = 0.594, sensitivity = 65.3%, specificity = 46.1%). PH-LHD was associated with increased HF hospitalization (HR = 2.344, *p* < 0.001) and mortality (HR = 2.015, *p* < 0.001) after adjusting for age and sex. In conclusion, persistent PH-LHD was found in 21.6% in the follow-up echocardiography and was associated with decreased PALS (<12.5%). PH-LHD persistence was associated with poor clinical outcomes. Thus, AHF patients with decreased PALS, especially <12.5%, should be followed with caution.

## 1. Introduction

Pulmonary hypertension (PH) associated with left heart disease (PH-LHD) is the most common form of PH and has significantly higher morbidity and mortality. Within the complex pathophysiology of PH-LHD, congestion and pulmonary vascular remodeling are the main contributing mechanism of PH-LHD [1,2,3]. The left atrium (LA) is an important chamber between the pulmonary artery and the left ventricle (LV), and it plays an essential role in the development of PH-LHD. Systemic congestion can increase left atrial (LA) pressure and volume and can be observed in many patients with significant valvular heart disease and heart failure (HF) [4]. Increased LA size and decreased LA compliance can be associated with increased LA pressure and contribute to PH-LHD development [5]. In addition, LA structural and functional change has also been associated with the development of HF [6].

Conventional echocardiographic assessment of the LA includes its size and volume, which does not represent LA intrinsic function. However, speckle-tracking echocardiography can measure LA strain, representing intrinsic function, with relatively high accuracy [7,8]. LA strain can be used in the detection of functional change of LA in its early stages and in the prediction of future development of atrial fibrillation (AF) and adverse clinical outcomes [8,9,10,11]. However, the effect of decreased LA strain on the persistence of PH-LHD is poorly studied. We evaluated the role of the LA intrinsic function on the persistent PH-LHD in acute HF (AHF) patients during follow-up echocardiographic examinations after stabilization.

## 2. Materials and Methods

### 2.1. Study Population

The STRATS-AHF (STrain for Risk Assessment and Therapeutic Strategies in patients with Acute Heart Failure) registry (NCT: 03513653, https://clinicaltrials.gov/ct2/show/NCT03513653, accessed on 1 March 2022) enrolled 4312 consecutive patients hospitalized with AHF at 3 tertiary university teaching hospitals in Korea from January 2009 through December 2016 [12]. We defined AHF as rapid-onset or worsening heart HF symptoms with or without objective signs needing urgent evaluation and management of HF [13]. All patients had symptoms or signs of pulmonary edema and either objective signs of structural heart disease or abnormal left ventricular (LV) function. The exclusion criteria were as follows: patients with acute coronary syndrome requiring surgery or those with severe primary valvular heart diseases; we also excluded patients without follow-up echocardiographic examination.

We checked all-cause deaths and HF admission in all patients using medical records and data from the Ministry of Public Administration and Security of the Republic of Korea. This study protocol was approved by each hospital’s institutional review board (IRB). The study complied with the Declaration of Helsinki principles.

### 2.2. Echocardiographic Examination

Initial and follow-up echocardiographic examinations were done with commercial echocardiographic machines (GE Vivid 9 and E90 (GE Vingmed Ultrasound AS, Horten, Norway), Philips iE33 (Philips Ultrasound, Bothell, WA, USA), and Acuson SC2000 (Siemens Medical Solutions USA, Inc., Mountain View, CA, USA)) and a 2.5 MHz probe using standard echocardiographic techniques suggested by the American Society of Echocardiography [14]. Echocardiographic modalities included M-mode, B-mode (two-dimensional echocardiographic mode), and Doppler mode. We measured LV end-systolic (LVESV) and end-diastolic volumes (LVEDV) with two-dimensional Simpson’s method from the apical 4- and 2-chamber views and calculated the LV ejection fraction (LVEF) with these values. LA anterior–posterior diameter was measured from the parasternal long-axis view. We checked mitral inflow velocity with the pulsed-wave Doppler echocardiographic modality, including E and A velocities, and mitral annular velocities with the tissue Doppler modality, including early-diastolic (E′), late-diastolic (A′), and peak systolic (S′) velocities. LV diastolic function was estimated with mitral E/A ratio, deceleration time, and mitral E/E’ ratio. The maximal velocity of tricuspid regurgitation (TR Vmax) was calculated with the continuous wave Doppler. Pulmonary artery systolic pressure (PASP) is similar to the systolic right ventricular pressure (RVSP) without significant pulmonary valve diseases. Therefore, we estimated PASP from the TR Vmax with this formula (PASP = 4 × TR Vmax^2^ + right atrial pressure). Because a TR Vmax of more than 3.4 m/s is considered to indicate a high probability of PH regardless of right atrial pressure, we defined the presence and persistence of PH with the TR Vmax of more than 3.4 m/s in the follow-up echocardiographic examinations [15]. Based on the baseline echocardiographic findings, patients were categorized as having HF with reduced ejection fraction (HFrEF, LVEF ≤ 40%), HF with mildly reduced ejection fraction (HFmrEF, LVEF between 41% and 49%), and HF with preserved ejection fraction (HFpEF, LVEF ≥ 50%) [16].

### 2.3. Strain Analysis

LA strain analysis is an assessment of LA myocardial deformation and can be measured by 2-dimensional speckle tracking, which tracks the speckle pattern frame by frame in standard 2-dimensional echocardiographic images [17]. Among several commercially available analyzing software, we measured LA strain values using TomTec-Arena version 4.6 (TomTec, Munich, Germany) from the stored initial echocardiographic images [18,19]. An echocardiographic specialist, unaware of the clinical data, calculated LA strain values in all participants independently with R–R gating as the zero reference point. After selecting suitable echocardiographic images for LA strain analysis from the echocardiographic database, the LA endocardial border was traced manually on the LV end-systolic frame. The LV end-systolic frame was defined from the electrocardiographic findings or the frame at the smallest LV volume during the cardiac cycle. Then, the software automatically tracks speckles along the endocardial border and myocardium throughout the cardiac cycle.

LA strains originally provides information about all 3 phases of LA function, including the reservoir, conduit, and contractile phases. Because 35.2% of our study population had AF, and LA conduit and contractile functions were not calculated, only the reservoir function was considered in this study. Peak atrial longitudinal strain (PALS), a marker of LA reservoir function, was calculated from the global longitudinal strain (GLS) average values from apical 4- and 2-chamber views. GLS values were calculated on a single cardiac cycle in patients with regular sinus rhythm, whereas they were calculated by an average of three cardiac cycles in those with AF and other arrhythmias. A higher PALS value represents a better LA reservoir function.

### 2.4. Statistical Analysis

We presented categorical variables as frequencies and continuous variables as means ± standard deviations. We divided our study population into two groups according to the presence of PH and performed Student’s *t*-test and χ^2^ test for categorical variables for comparisons between two groups. Because the time intervals from the initial echocardiography to the follow-up echocardiography were different, we adjusted the time intervals with Cox proportional hazard analysis to predict PH. We performed the multivariable analysis with the significant parameters on the univariate analysis. We included the most significant variable among the variables with multicollinearity with others in the multivariate analysis to avoid the overfitting of the model. Cox proportional hazard analysis was also applied to find the effect of PH on all-cause mortality or adverse clinical events. The data were analyzed using SPSS version 22 (IBM, Chicago, IL, USA) and MedCalc version 12.3.0.0 (MedCalc Software, Mariakerke, Belgium). We considered a variable with a two-sided *p*-value of <0.05 to be statistically significant.

## 3. Results

### 3.1. Patient Characteristics

We included 4312 patients in the STRAT-AHF cohort and excluded 547 patients with insufficient baseline echocardiographic examinations in this study. Patients without follow-up echocardiographic examinations were also excluded. Thus, we evaluated 1729 patients with follow-up echocardiographic images (Figure 1).

Between the initial and follow-up echocardiographic examinations, the mean duration was 18.1 ± 13.5 months. The mean patient age was 70.1 ± 14.2 years, 52.1% were men, and the mean LVEF was 39.4% ± 15.6% (Table 1).

Hypertension was the most common comorbidity (57.3%), and HFrEF, HFmrEF, and HFpEF were found in 919 patients (53.2%), 263 (15.2%), and 547 (31.6%), respectively. The mean LA diameter was 42.7 ± 8.2 mm, the mitral E/E′ ratio was 18.8 ± 11.0, the baseline TR Vmax was 3.0 ± 0.6 m/s, and the PALS was 14.7% ± 10.1%. PALS showed significant negative correlation with PASP (r = −0.190, *p* < 0.001).

We categorized the patients as either having no PH (n = 1356, 78.4%) or PH (n = 373, 21.6%) on follow-up echocardiography. Among 373 patients with persistent PH-LHD, HFrEF was found in 174 patients (46.6%), HFmrEF in 63 patients (17.4%), and HFpEF in 136 patients (37.6%). The PH group had significantly higher age, blood urea nitrogen (BUN), glucose, lower cholesterol, triglyceride, hemoglobin, and diastolic blood pressure (DBP). The prevalence of AF, hypertension, diabetes, HFpEF, and diuretic use at discharge was higher in the PH group. Baseline LVESV and PALS were lower. LA diameter, mitral E/E′ ratio, and TR Vmax were higher in the PH group.

### 3.2. Factors in the Prediction of Persistent PH in the Follow-Up Echocardiography

The univariate and multivariate analyses are summarized in Table 2.

Persistent PH was associated with systolic BP (HR = 0.995, *p* = 0.010), DBP (HR = 0.991, *p* = 0.004), diabetes (HR = 1.290, *p* = 0.017), total cholesterol (HR = 0.995, *p* < 0.001), triglyceride (HR = 0.997, *p* = 0.033), hemoglobin (HR = 0.926, *p* = 0.008), BUN (HR = 1.006, *p* = 0.014), glucose (HR = 1.002, *p* = 0.001), diuretic use (HR = 1.629, *p* < 0.001), LA diameter (HR = 1.021, *p* < 0.001), mitral E/E′ ratio (HR = 1.019, *p* < 0.001), TR Vmax (HR = 1.261, *p* < 0.001), and PALS (HR = 1.046, *p* < 0.001) in the univariate analysis. In the multivariate analysis, hemoglobin (HR = 0.925, *p* = 0.044), TR Vmax (HR = 2.017, *p* < 0.001), and PALS (HR = 1.024, *p* = 0.040) were significant variables after adjusting for other significant variables in the univariate analysis. Moreover, PALS showed statistical significance after adjusting for the LA diameter and the mitral E/E’ ratio. We analyzed a receiver operating curve analysis to find the best cutoff value to predict persistent PH and found PALS < 12.5% to be the best cutoff point (area under the curve = 0.594, sensitivity = 65.3%, specificity = 46.1%).

### 3.3. Effect of PH-LHD on the Clinical Outcome

We evaluated the effect of persistent PH-LHD on survival. The mean duration from the follow-up echocardiographic examinations to the last follow up was 44.0 ± 34.1 months. During the period, 583 patients died, and 399 patients were admitted for the aggravation of HF. The presence of PH-LHD was significantly associated with HF hospitalization (HR = 2.344, 95% confidence interval = 1.914–2.871, *p* < 0.001) and all-cause mortality (HR = 2.015, 95% confidence interval = 1.691–2.400, *p* < 0.001) after adjustment for age and sex (Figure 2).

## 4. Discussion

In our cohort composed of hospitalized patients for AHF, persistent PH-LHD was found in 21.6% in the follow-up echocardiography. After adjusting statistically significant variables, including LA diameter and mitral E/E’ ratio, PALS was a significant determinant of PH in the follow-up echocardiographic examinations. The best cutoff value of PALS in predicting the persistent PH-LHD was less than 12.5%. Persistent PH-LHD was also a significant prognostic factor of all-cause mortality and HF hospitalization after adjustment for age and sex.

PH-LHD is the most common form of PH, and there are three main entities in PH-LHD, including PH due to HFpEF, PH due to HFrEF, and PH due to valvular heart disease (VHD) [15]. In our study composed of hospitalized AHF patients, PH-LHD was most commonly associated with HFrEF (45%). PH is a pathophysiologically relevant phenomenon and common accompanying feature in HF, even in optimally treated patients [1]. Reduced pulmonary artery compliance and increased pulmonary artery pressure occur as a consequence of the increase in LV filling pressure and pulmonary capillary wedge pressure [20]. Thus, PH-LHD could result from various cardiac disorders that increase LA pressure [21]. Because LA plays a central role in the modulation and optimization of LV filling and cardiac output, LA function and LV filling pressure are closely related. The increased LV filling pressure is transmitted back into the LA and increases LA pressure and wall stress, eventually, causing LA dysfunction. The decreased LA function further aggravates LV filling pressure and cannot compensate for LV diastolic dysfunction and can also lead to deficient atrial natriuretic peptide synthesis and regulation, which contribute to further elevation of pressure in LA and pulmonary circulation [22]. Furthermore, chronically elevated LA pressure can alter the pulmonary arterial structure, and elevated pulmonary vascular resistance can be observed in the combined post and precapillary pulmonary hypertension [23].

Echocardiographic parameters that can be used to predict PH-LHD include increased LA size (LA anteroposterior diameter > 4.2 cm), increased LA pressure assessed by increased mitral E/E’ ratio (mitral E/E′ ratio > 10), increased right ventricular cavity size, and increased LV mass index [24,25,26,27]. Although PALS can be influenced by preload (LA volume), afterload (E/E′ ratio), LV systolic function, and extent of LA fibrosis [28], we showed that PALS was a significant variable in predicting persistent PH-LHD after adjusting for LA size and mitral E/E’ ratio in this study.

LA strain is a simple and reproducible marker of phasic LA function and has additional prognostic value over conventional echocardiographic parameters. LA strain correlates with increasing severity of LV diastolic dysfunction in patients with preserved LVEF and is useful to differentiate diastolic dysfunction [29,30,31]. Potter et al. published that replacing an LA volume index of 34 mL/m^2^ with a PALS of <24% can re-categorize diastolic function in 75% of cases with indeterminate diastolic dysfunction [32]. Significant diastolic dysfunction categorized by LA strain was independently associated with incident HF in this study. Decreased PALS was associated with the development of future AF and stroke events in AHF patients [10,33,34,35]. Decreased PALS is also associated with increased LA fibrosis and LA stiffness [35,36]. In a study with pathologic examination of the resected LA during heart transplantation, PALS showed a better correlation with LA fibrosis (r = −0.88, *p* < 0.001) than with mitral E/E′ ratio (r = 0.44, *p* = 0.005) in patients with advanced HF [37].

In addition, PALS is a reliable and feasible value on top of conventional parameters to characterize HF patients and predict prognosis [12,38,39,40]. Rossi et al. reported that decreased PALS (≤23%) was associated with overall mortality or HF hospitalization regardless of LA size in 626 patients with HFrEF [38]. PALS showed a significant correlation with H2FPEF score in 1105 patients with HFpEF, irrespective of the presence of AF [39]. Thus, PALS has become an essential echocardiographic parameter in evaluating HFpEF patients because of its strong correlation with invasive LV filling pressure and excellent feasibility [40].

The continuous increase in LA pressure leads to persistent PH, resulting in functional impairment and ultrastructural change of pulmonary vasculature. We showed that the presence of PH-LHD in the follow-up echocardiographic examinations was associated with increased HF hospitalization and mortality. These findings were consistent with previous studies showing that PH-LHD was a significant predictor of worse clinical outcomes [41,42,43]. Though the reversibility of PH-LHD has not been fully known, pulmonary-artery-specific vasodilators can be applied in a specific subset of patients with persistent PH-LHD and pre-capillary component [15,23]. Though the diagnostic gold standard of PH-LHD is right heart catheterization (RHC), given the absence of proven treatments for PH-LHD, it is impossible to perform RHC in all AHF patients. If we performed RHC to confirm the PH with a pre-capillary component, we could identify patients who would benefit from vasodilators and might increase their survival rate.

Because decreased PALS was associated with the persistence of PH in this study, patients with decreased PALS, especially PALS < 12.5%, might undergo RHC to assess the possibility of combined pre- and postcapillary PH. This strategy might guide the management of hidden PH-LHD and improve survival. However, more investigations about this topic are needed at this time.

### Limitations

This study had several limitations. First, this study was a retrospective study with reanalysis of stored echocardiographic images obtained using commercially available echocardiographic machines. Although we used a vendor-independent algorithm in measuring PALS (TomTec-Arena version 4.6) with good reproducibility [44], there might be vendor-based differences among echocardiographic machines. Second, the intervals between the baseline echocardiography and the follow-up echocardiography were variable in our study population. Therefore, we adjusted the time interval with the Cox proportional hazard analysis to predict persistent PH-LHD. Third, the diagnosis of PH by echocardiographic examinations is not the standard [44]. However, TR Vmax assessed by echocardiography is the most commonly used screening tool [15]. TR Vmax can be influenced by many other factors. Thus, we defined the presence of PH with TR Vmax ≥ 3.4 m/s to increase the diagnostic accuracy of detecting PH.

To confirm the effect of decreased PALS on the persistent PH-LHD, we should have a prospective study with regular echocardiographic follow ups.

## 5. Conclusions

Persistent PH-LHD was found in 21.6% in the follow-up echocardiography in our STRAT-AHF cohort. PALS was a significant determinant in the predicting PH-LHD after adjusting for LA diameter and mitral E/E′ ratio, and the best cutoff was PALS < 12.5%. Moreover, the presence of PH-LHD was a poor prognostic factor in our study population.

## Figures and Tables

**Figure 1 jcm-11-03510-f001:**
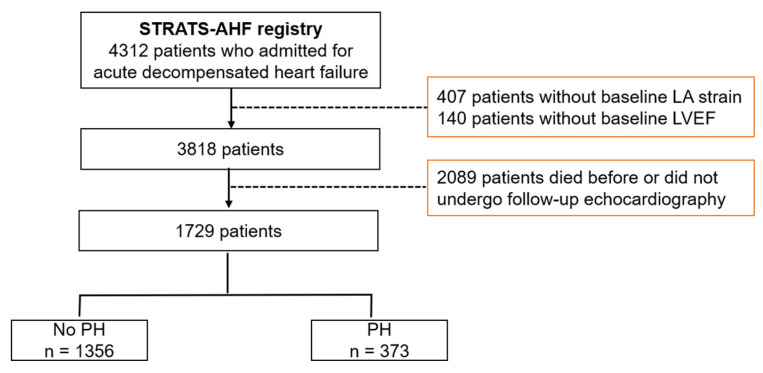
Study scheme. LA: left atrium, LVEF: left ventricular ejection fraction, PH: pulmonary hypertension, STRATS-AHF: STrain for Risk Assessment and Therapeutic Strategies in patients with Acute Heart Failure.

**Figure 2 jcm-11-03510-f002:**
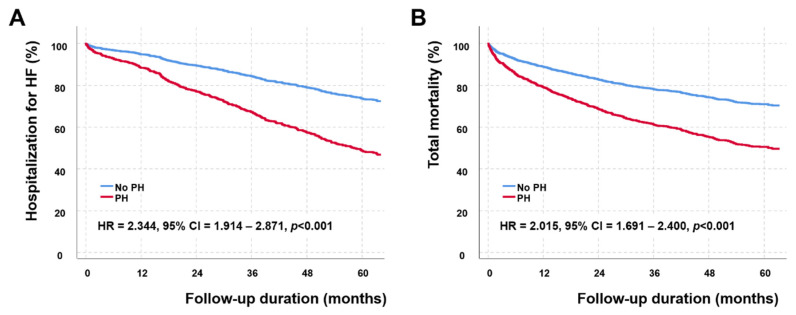
Event-free survival (**A**) and total survival (**B**) curves according to the presence of pulmonary hypertension associated with left heart disease (PH-LHD). Patients with PH-LHD have significantly higher hospitalization for heart failure (hazard ratio (HR) = 2.344, 95% confidence interval (CI) = 1.914–2.871, *p* < 0.001) and all-cause mortality (HR = 2.015, 95% CI = 1.691–2.400, *p* < 0.001) assessed by the Cox proportional hazard analysis after adjusting for age and sex.

**Table 1 jcm-11-03510-t001:** Comparison of baseline clinical characteristics and echocardiographic data according to the presence of pulmonary hypertension at follow-up echocardiography.

	Total (n = 1729)	No PH (n = 1356)	PH (n = 373)	*p* Value
Male (%)	900 (52.1)	713 (52.6)	187 (50.1)	0.413
Age (years)	70.1 ± 14.2	68.5 ± 13.5	71.3 ± 12.4	<0.001
BMI (kg/m^2^)	23.3 ± 4.1	23.7 ± 4.0	23.5 ± 4.0	0.306
NYHA Fc IV (%)	686 (44.6)	536 (44.9)	150 (43.4)	0.623
AF (%)	605 (35.2)	447 (33.2)	158 (42.6)	0.001
**Physical examination**				
SBP (mmHg)	129.7 ± 27.5	129.5 ± 26.5	126.9 ± 26.8	0.092
DBP (mmHg)	73.9 ± 16.5	75.4 ± 16.8	72.7 ± 16.3	0.006
Heart rate (bpm)	85.4 ± 22.7	88.6 ± 25.5	87.8 ± 24.8	0.594
**Past medical history**				
Hypertension (%)	991 (57.3)	756 (55.8)	235 (63.0)	0.013
Diabetes (%)	564 (32.6)	422 (31.1)	142 (38.1)	0.013
Ischemic heart disease (%)	521 (30.1)	407 (30.0%)	114 (30.6)	0.849
**Laboratory finding**				
Total cholesterol (mg/dL)	157.8 ± 44.6	156.0 ± 42.7	145.9 ± 43.3	<0.001
Triglyceride (mg/dL)	113.7 ± 75.4	113.3 ± 80.3	99.4 ± 57.2	0.006
HDL-cholesterol (mg/dL)	43.5 ± 13.4	44.0 ± 13.0	45.9 ± 13.8	0.067
Hb (g/dL)	12.0 ± 2.3	12.4 ± 2.3	11.9 ± 2.3	0.004
BUN (mg/dL)	26.3 ± 17.0	24.5 ± 15.7	28.2 ± 17.5	<0.001
Cr (mg/dL)	1.7 ± 2.0	1.5 ± 1.8	1.7 ± 2.0	0.175
Glucose (mg/dL)	156.0 ± 78.5	143.1 ± 69.3	155.7 ± 75.1	0.004
NT proBNP (pg/mL)	7962 ± 11,305	7656 ± 11,149	9026 ± 12,011	0.095
**Baseline echocardiographic finding**				
LVEDD (mm)	53.9 ± 9.6	54.3 ± 9.4	53.9 ± 9.3	0.461
LVESD (mm)	41.6 ± 11.8	41.6 ± 11.6	40.3 ± 11.3	0.070
LVEDV (mL)	127.7 ± 65.1	125.8 ± 65.4	118.8 ± 60.2	0.091
LVESV (mL)	84.1 ± 57.0	81.8 ± 56.2	74.0 ± 52.0	0.026
LVEF (%)	39.4 ± 15.6	40.2 ± 15.3	42.7 ± 15.5	0.006
LVGLS (%)	−11.0 ± 4.9	−11.1 ± 4.9	−10.7 ± 4.8	0.135
LA diameter (mm)	42.7 ± 8.2	45.3 ± 9.1	49.2 ± 11.0	<0.001
LAVI (mL/m^2^)	63.7 ± 42.1	59.6 ± 34.4	78.4 ± 60.5	<0.001
Mitral E/E’ ratio	18.8 ± 11.0	18.2 ± 9.5	21.1 ± 11.8	<0.001
TR Vmax (m/s)	3.0 ± 0.6	2.9 ± 0.7	3.2 ± 0.6	<0.001
PALS (%)	17.0 ± 10.5	13.9 ± 9.5	11.5 ± 7.8	<0.001
**Phenotype of HF (%)**				0.017
HFrEF (%)	919 (53.2)	745 (54.9)	174 (46.6)	
HFmrEF (%)	263 (15.2)	200 (14.7)	63 (17.4)	
HFpEF (%)	547 (31.6)	411 (30.3)	136 (37.6)	
**Medication at discharge**				
Beta-blocker (%)	1152 (66.9)	909 (67.2)	243 (65.5)	0.534
RAS blocker (%)	1299 (75.4)	1017 (75.2)	282 (76.0)	0.786
MRA (%)	876 (50.8)	680 (50.3)	196 (52.8)	0.412
Diuretics (%)	1344 (78.0)	1037 (76.7)	307 (82.7)	0.013

AF: atrial fibrillation, BMI: body mass index, BUN: blood urea nitrogen, Cr: creatinine, DBP: diastolic blood pressure, Hb: hemoglobin, HDL-cholesterol: high-density-lipoprotein cholesterol, HFrEF: heart failure with reduced ejection fraction, HFmrEF: heart failure with mildly reduced ejection fraction, HFpEF: heart failure with preserved ejection fraction, NYHA Fc: New York Heart Association functional class, NT proBNP: N-terminal pro B-type natriuretic peptide, LA: left atrium, LAVI: left atrial volume index, LVEDD: left ventricular end-diastolic dimension, LVESD: left ventricular end-systolic dimension, LVEDV: left ventricular end-diastolic volume, LVESV: left ventricular end-systolic volume, LVEF: left ventricular ejection fraction, LVGLS: left ventricular global peak systolic longitudinal strain, MRA: mineralocorticoid antagonist, NYHA: New York Heart Association, PALS: peak atrial longitudinal strain, PH: pulmonary hypertension, RAS: renin–angiotensin–aldosterone system, SBP: systolic blood pressure, TR Vmax: maximal velocity of tricuspid regurgitation.

**Table 2 jcm-11-03510-t002:** Univariate and multivariate analysis of the prediction of pulmonary hypertension at follow-up echocardiography.

Variable	HR	95% CI	*p* Value
**Univariate analysis**			
Male	0.956	0.780–1.172	0.666
Age (years)	1.006	0.998–1.015	0.135
BMI (kg/m^2^)	0.982	0.957–1.008	0.173
NYHA Fc IV	0.844	0.681–1.046	0.122
AF	1.098	0.893–1.350	0.374
SBP (mmHg)	0.995	0.991–0.999	0.010
DBP (mmHg)	0.991	0.985–0.997	0.004
Heart rate (bpm)	1.000	0.996–1.004	0.914
Hypertension	1.174	0.951–1.449	0.135
Diabetes	1.290	1.046–1.590	0.017
Ischemic heart disease	0.995	0.798–1.240	0.961
Total cholesterol (mg/dL)	0.995	0.993–0.998	<0.001
Triglyceride (mg/dL)	0.997	0.995–1.000	0.033
HDL-cholesterol (mg/dL)	1.008	0.997–1.018	0.164
Hb (g/dL)	0.926	0.875–0.980	0.008
BUN (mg/dL)	1.006	1.001–1.011	0.014
Cr (mg/dL)	1.025	0.976–1.077	0.317
Glucose (mg/dL)	1.002	1.001–1.003	0.001
LVEDD (mm)	1.002	0.990–1.013	0.764
LVESD (mm)	1.003	0.994–1.013	0.500
LVEDV (mL)	1.000	0.999–1.002	0.605
LVESV (mL)	1.000	0.998–1.003	0.658
LVEF (%)	0.996	0.989–1.003	0.256
LA diameter (mm)	1.021	1.011–1.031	<0.001
Mitral E/E’ ratio	1.019	1.010–1.027	<0.001
TR Vmax (m/s)	1.261	1.167–1.362	<0.001
PALS (per 1% decrease)	1.046	1.020–1.049	<0.001
Phenotype of HF			
HFrEF	Reference		
HFmrEF	1.022	0.764–1.368	0.881
HFpEF	0.944	0.750–1.187	0.622
Beta-blockers at discharge	1.099	0.887–1.362	0.387
RAS blockers at discharge	1.194	0.940–1.516	0.146
MRA at discharge	0.833	0.679–1.021	0.079
Diuretics at discharge	1.629	1.244–2.133	<0.001
**Multivariate analysis**			
DBP (mmHg)	0.991	0.982–1.003	0.063
Diabetes	1.333	0.999–1.947	0.078
Total cholesterol (mg/dL)	1.000	0.996–1.004	0.959
Hb (g/dL)	0.925	0.857–1.998	0.044
BUN (mg/dL)	1.003	0.995–1.010	0.496
LA diameter (mm)	1.007	0.989–1.024	0.450
Mitral E/E’ ratio	1.000	0.988–1.013	0.942
TR Vmax (m/s)	2.017	1.517–2.683	<0.001
MRA at discharge	1.070	0.768–1.490	0.690
Diuretics at discharge	1.309	0.860–1.991	0.209
PALS (per 1% decrease)	1.024	1.001–1.048	0.040

AF: atrial fibrillation, BMI: body mass index, BUN: blood urea nitrogen, CI: confidence interval, Cr: creatinine, DBP: diastolic blood pressure, Hb: hemoglobin, HDL-cholesterol: high-density-lipoprotein cholesterol, HFrEF: heart failure with reduced ejection fraction, HFmrEF: heart failure with mildly reduced ejection fraction, HFpEF: heart failure with preserved ejection fraction, HR: hazard ratio, NYHA Fc: New York Heart Association functional class, LA: left atrium, LVEDD: left ventricular end-diastolic dimension, LVESD: left ventricular end-systolic dimension, LVEDV: left ventricular end-diastolic volume, LVESV: left ventricular end-systolic volume, LVEF: left ventricular ejection fraction, MRA: mineralocorticoid antagonist, NYHA: New York Heart Association, PALS: peak atrial longitudinal strain, PH: pulmonary hypertension, RAS: renin–angiotensin–aldosterone system, SBP: systolic blood pressure, TR Vmax: maximal velocity of tricuspid regurgitation.

## Data Availability

Not applicable.

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
