# Peer review of "Decreased Peak Left Atrial Longitudinal Strain Is Associated with Persistent Pulmonary Hypertension Associated with Left Heart Disease"

_jcm, 2022, doi:10.3390/jcm11123510_

Round 1

Reviewer 1 Report

This is an interesting study investigating correlation between PALS and PH-LHD in a large group of patients with acute heart failure. The study seems to be conducted and planned well, the topic is interesting. 

My major concern is diagnosis of PH. The authors should:

1) clearly define how they diagnosed PH

2) attempt to estimate pulmonary artery pressure (PAP) / right ventricular pressure from their echo data (formulas are widely available and used)

3) correlate PAP with their PALS  results.

PALS should be described in more detail, since it is the key parameter in the study, i.e. what it actually measures.   

Author Response

Thank you for your comments, and we have sincerely responded to your comments. Your comment make our manuscript better. Thank you again,

Reviewer 2 Report

The authors conducted an interesting analysis of more than a thousand patients hospitalized for acute HF, showing a potential value of left atrial strain, beyond ejection fraction, as a predictor of persistent pulmonary hypertension at follow up, which was also correlated with poorer prognosis.

The big cohort and the used of advanced echocardiographic parameters are two advantages of the study.

However, I think that the main objectives and results of the study should be better highlighted by the authors. Moreover,there are several lacking information and old indexed used instead of superior parameters that are widely validated and recommended. AUC showing the predictive value of LA strain is not good.

However, the results are interesting as preliminary results for hypothesis generation.

Therefore, I suggest these major and minor revisions: 

Major revisions:

The authors should justify their choice to define persistent PH based on tricuspid regurgitant velocity. The guidelines to which they refer, in fact, define PH based on clinical and instrumental findings, and concerning the recommended pulmonary pressure estimation it states that  “The estimation of systolic PAP is based on the peak tricuspid regurgitation velocity (TRV) taking into account right atrial pressure (RAP) as described by the simplified Bernoulli equation. “

Did the authors also measure left ventricular global longitudinal strain? If not,why? Since it may have offered additive information on subtle left ventricular dysfunction beyond LV ejection fraction [PMID: 29413646.]. Please discuss it 

Did the authors measure NTproBNP or BNP? This is an essential marker in heart failure patients and I think that it should have been measured at least at hospital admission for acute HF. If available, please provide these values in the PH/noPH groups

Are LA volume (better if indexed for BSA) available? Since we know that LA diameter offer an estiation of LA dimensions in only one dimension, LA volume is a more reliable measure to estimate LA dilatation.

Please add a table showing population characteristics in the three groups (HFrEF, HFmrEF, HFpEF)

Some clinical and echocardiographic characteristics at follow up should be provided in order to link the persistence of pulmonary hypertension with symptoms and signs of congestion.

The mechanism of pulmonary hypertension link with heart failure should be better explained in the introduction or in the discussion section [see: PMID: 31025236, PMID: 32969856 ]

The role of left atrial strain in the three clinical settings mentioned in line 214 [HFpEF, HFrEF, valvular heart disease, see: PMID: 34726345 , PMID: 33035685, PMID: 34729586, PMID: 35014120] and its additional prognostic value over baseline echocardiographic parameters should be more extensively discussed to inform the readers about the main study topic.

The clinical impact of study results should be better explained to focus the main results of the study. For example, when authors ended the discussion mentioning the role of assessing PH with right heart catheterization (RHC) to improve PH management and survival, they should link it with a possible role of LA strain as a surrogate of RHC to guide the assessment of persistent PH, as they demonstrated that LA strain was a predictor of this clinical condition.

Minor comments:

Abstract should report the conclusions, in means of clinical impact, of the study

Line 51 , I suggest to change “the effect of LA stiffness assessed by LA strain” if it is not referred to the parameter “LA stiffness” obtained as E/E’ / PALS ratio (es. PMID: 31093897), since it may be confusing.

Heart failure guidelines [ref.13] should be updated to the latest version of the guidelines 

Please specify the echocardiographic machine/s used for echocardiographic exam 

Line 207: the discussion begin should also mention the kind of study population analyzed (patients hospitalized for acute HF?)

Author Response

(The authors gave the same response as above.)

Round 2

Reviewer 1 Report

I am satisifed with the authors' response